# Spatio-temporal evolution of habitat quality and its influencing factors in karst areas based on the InVEST model

**Chao Ma** ⓘ *, **Huituo Yang, Zhi Yan**

College of Materials Science and Engineering, Institution, Guiyang Univiersity, Guiyang , China

* 18685167041@163.com

## Abstract

The Critical Karst Zone provides rich natural resources and is an important habitat for the survival and development of the world's human population. Meanwhile, urbanization processes have disrupted the structure and function of natural ecosystems, endangering biodiversity and habitats. However, existing studies have few frequently explored the combined effects of the natural environment and human activities on changes in habitat quality. This article uses the InVEST model and the GeoDetector method to analyze the changes in landscape patterns, spatiotemporal evolution of habitat quality, and their driving factors in karst areas. The results show that: (i) From 1990 to 2020, forest, cropland, and grassland fluctuated sharply, while the building and waters area showed an exponential upward trend. The overall landscape fragmentation and spatial heterogeneity are enhanced. (ii) The karst habitat quality index decreased from 0.7751 to 0.74085, showing an overall downward trend. The habitat quality shows a spatial distribution pattern of "high in the surrounding areas and low in the central areas", and autocorrelation analysis shows that county-level units have significant spatial agglomeration effects. (iii) The overall type shows an enhancement of dual factor or non-linear, in which land use intensity and population density are the main driving factors for the spatio-temporal evolution of habitat quality. In summary, adopting stringent ecological protection and restoration initiatives aimed at minimizing human activity intensity and safeguarding natural habitat integrity in karst regions is imperative. Such measures contribute to the scientific underpinning for decision-making regarding the optimization of regional landscape composition and enhance land spatial planning strategies.

## 1 Introduction

Habitat quality(HQ) is the ability of the ecological environment to provide suitable conditions for the survival and development of organisms, as well as an important representation of regional ecosystem function and biodiversity [1]. Since the Industrial Revolution, high-intensity human socio-economic activities have produced many drawbacks. For example, climate change, resource depletion, and increased urbanization [2]. The Millennium Ecosystem Assessment report indicates that 60% of all types of ecosystems worldwide have been severely

**Data availability statement:** All relevant data are within the paper and its Supporting Information files

**Funding:** This research was funded by Guiyang science and technology plan project (Zhuke contract [2021] No.43-17. Guiyang University introduced talents to start the funds for scientific research project GYU-KYU. Guiyang University introduced talents to start the funds for scientific research project GYU-KY- [2025]. The funder of this study, Chao Ma, was responsible for the concept, the design, the writing and the financial support.

**Competing interests:** The authors have declared that no competing interests exist.

degraded [3]. The large-scale transformation of forests and grasslands into cropland and buildings has altered the structure and function of ecosystems, leading to a decline in global biodiversity [4–6]. In addition, the Life on Earth Report states that species populations have declined by more than 50% globally [7]. The loss and destruction of habitats have become a worldwide issue. To prevent and reverse the damage to human well-being caused by the degradation of HQ, the United Nations Convention on Biological Diversity pointed out that all humankind should take responsibility for identifying and monitoring important biological components, establishing nature reserves, and promoting the restoration of biodiversity [8,9]. However, we first need to understand and clarify the characteristics of methods and processes for assessing HQ at different stages, which is beneficial for providing solutions for the sustainable use of land resources.

Habitat quality is an important basis for maintaining ecosystem functions and providing ecosystem services. Current research has classified HQ assessment methods into three categories, namely field surveys, biophysical indicators, and ecological modeling, based on differences in subject, content, and scale [10]. Field survey assessment focuses on the biodiversity of regional populations and communities, as well as niche surveys. The method is only suitable for static assessments of small areas and short time series [11]. Biophysical indicators are based on remote sensing to provide an integrated assessment of ecological indicators, such as vegetation cover and net primary productivity (NPP). Boundary identification, data normalization, and indicator weighting are prerequisites of the approach, which lacks uniformity and generalizability [12]. However, the development of "3S" technology offers the possibility of assessing the spatial and temporal dynamics of ecosystems across multiple scales and long time series [13]. Examples include the Integrated Valuation and Trade-off of Ecosystem Services (InVEST) model and the Social Value of Ecosystem Services (SoLVES) model. Among them, the SoLVES model will integrate environmental and social values, but it is difficult and contradictory to quantify the aspects of values, attitudes, and preferences of different stakeholder subjects [14,15]. However, the InVEST model, on the other hand, integrates habitat adaptations and levels of human disturbance and has the advantages of simplicity of use, easy access to data, and strong spatial visualization [16]. As a result, the InVEST model has been widely used to study HQ restoration strategies at large scales such as in watersheds, provinces, and globally [17–19], especially in studies based on the effects of land use change on HQ and biodiversity [20]. However, a single land use does not capture the extent to which human activities interfere with complex niche ecosystem processes. This is particularly true in karst areas, where environmental fragility, vegetation degradation, and soil erosion are common phenomena. The landscape pattern index has the advantage of quantitatively analyzing the degree of fragmentation, separation, biodiversity, and disturbance of the spatial structure [21]. The evaluation method combining the InVEST model and the landscape pattern index can provide new ideas for the study of ecological status, spatial heterogeneity, and human activities in karst areas.

Karst areas account for only about 15% of the world's land area but provide freshwater resources for about 20-25% of humanity [22]. The South China Karst is the world's most typical, extensive, and geomorphologic type-rich ecological fragile area in the world, providing rich raw materials, biodiversity, culture, and other resources [23]. Due to the unique binary three-dimensional geomorphic structure and hydrologic processes of karst, a large amount of $CO_2$-laden rainwater or snowmelt infiltrates vertically. Carbonate rocks (dolomite and limestone) of shallow karst fractures have undergone severe chemical dissolution, resulting in slow soil formation rates and severe nutrient loss [24]. The phenomenon of large-scale exposed bedrock, discontinuous soil layers, and severe degradation of land productivity occurs in the Epikarst Zone [25]. To curb the expansion of karst desertification, China has implemented a series

of ecological restoration projects. As an ecological barrier in the upper reaches of the Yangtze River and the Pearl River, the HQ of Guizhou has related to the ecological security of the Yangtze River basin. Therefore, it is urgent to clarify the spatio-temporal evolution and driving factors of HQ before and after the implementation of karst desertification control projects.

Taking Guizhou Province as the research object, this study aims to explore the changing pattern of karst landscape pattern, the spatial and temporal evolution of HQ, and driving factors in different periods. Here, we assume that: (i) With the increase of human activities, karst areas experience severe land use transfer and their landscape fragmentation and heterogeneity increase. (ii) With the increase in land use intensity, the HQ at different stages shows a downward trend. (iii) Due to the fragile geological landscape, low ecological carrying capacity, and insufficient service capacity of karst areas, once the density of the impoverished population exceeds the ecological threshold, the ecological quality will be seriously threatened. Therefore, land use intensity and population density are the main factors limiting the recovery of karst ecosystems. To verify these hypotheses, we quantified the HQ of concentrated and contiguous karst areas worldwide. Systematically explain the relationships between HQ, land use intensity, population density, and ecological management projects.

## 2 Materials and methods

### 2.1. Study area

Guizhou Province (103°36′-109°35′E, 24°37′-29°13′N) is located in the southwest of the Yunnan-Guizhou Plateau, with an area of about $1.76 \times 10^5$ km² (Fig 1). Among them, the total exposed area of carbonate rocks is about $1.09 \times 10^5$ km², accounting for 62.13% of the total area of the region [26]. Due to its unique surface and subsurface binary

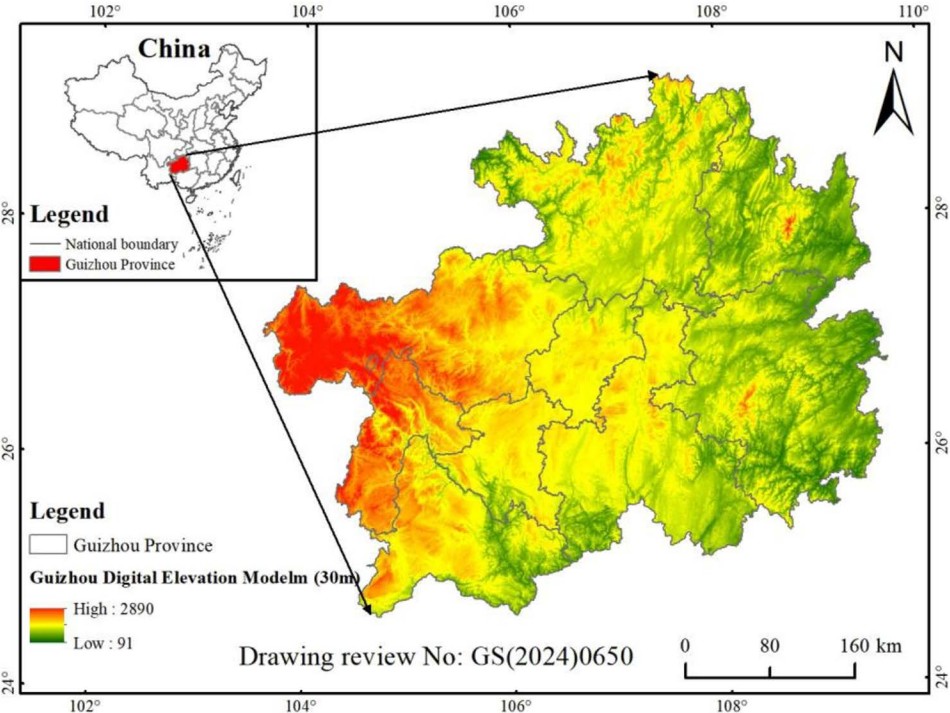

**Fig 1. Location map of the study area.**

three-dimensional structure landform structure and inappropriate economic activities in the region, its ecosystem is extremely fragile and has poor stability. The landform is mainly composed of plateau mountains, hills, and basins, with a terrain higher in the west and lower in the east, with an average altitude of about 1100m. Belonging to the subtropical monsoon humid climate zone, the annual average temperature is about 16.2°C, and the annual average precipitation is about 1013.6 mm. The main types of soil are yellow soil, lime soil, yellow -brown soil, red soil, brown soil, and purple soil. The vegetation is mainly composed of subtropical evergreen broad-leaved forests, and coniferous and broad-leaved mixed forests [27].

## 2.2.  Research framework

Firstly, we used ArcGis10.2 software to process land use data from 1990 to 2020. On this basis, we extracted the landscape pattern index. Then, we ran the InVEST model and GeoDa 1.22 software to obtain spatio-temporal variation data of HQ. Finally, combining data (climate data, DEM, and socio-economic), the Geodatector model was run to explore the driving factor of the spatial and temporal evolution of HQ (Fig 2).

## 2.3.  Data source

The land use data for the four periods (1990, 2000, 2010, and 2020) presented in this article were all from a 30 m × 30 m resolution raster dataset provided by the Resource and Environment Science and Data Center (http://www.resdc.cn). The meteorological data comes from the Resource and Environment Science and Data Center (http://www.resdc.cn). The Digital Elevation Model (DEM) data comes from the Geospatial Data Cloud (http://www.gscloud.cn). The relevant data sources are listed below (Table 1).

## 2.4.  Study methods

### 2.4.1.  Land use transfer.  The land use transfer matrix revealed the sources and composition methods of land types across different periods [28]. These raster data were

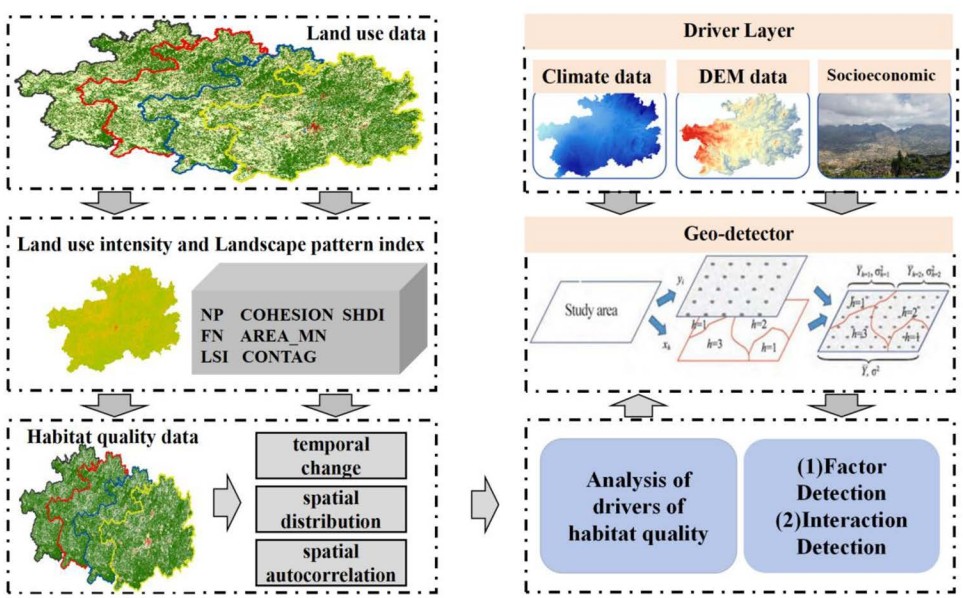

**Fig 2.  Research Technology Flow Chart.**

reclassified into six categories: cropland, forest land, grassland, waters, building, and unused land. And construct a transition matrix by spatial functions with the following conversion formula [29]:

$$N = 10A + B \tag{1}$$

where, $N$ is the unit of variation. $A$ is the initial unit of study. $B$ is the final unit of study.

**2.4.2. Landscape pattern changes.** Karst was characterized by high landscape fragmentation and spatial heterogeneity strength, which affect the landscape pattern to varying degrees. Therefore, we selected seven indicators, including the number of patches, landscape shape index, and fragmentation index, and used Fragstats 4.2.1 to analyze the characteristics and evolution trends of the landscape pattern index in the study area [30]. The landscape index and their ecological significance are put in the supplementary materials S1.

**2.4.3. Land use intensity.** Land use intensity was the degree of impact of human survival and development on natural ecosystems, and its essence is the relationship between human society and land supply and demand [31]. At the same time, according to the classification criteria, the land use intensity type was classified into four levels in combination with the actual study area [32] (Table 2). Because the spatial auto-correlation and driver analysis in the paper were based on county units. Therefore, the land use intensity used an average of county units. The formula is as follows:

$$L = 100 \sum_{i=1}^{n} J_i \times A_i \, , \, L \in \left[100 \, , \, 400\right] \tag{2}$$

**Table 1. Data source.**

| Name | Type | Data source | Note | Time |
|---|---|---|---|---|
| **Administrative boundary** | EPS | Standard map service http://bzdt.ch.mnr.gov.cn/index.html | no | Accessed on 10 October 2023 |
| **Land use/Land cover** | Raster | Resource and Environment Science and Data Center http://www.resdc.cn | 30m × 30m | |
| **Annual temperature** | Raster | China Meteorological Data Center, http://data.cma.cn/ | 1000m × 1000m | |
| **Annual precipitation** | Raster | China Meteorological Data Center, http://data.cma.cn/ | 1000m × 1000m | |
| **NDVI** | Raster | Resource and Environment Science and Data Center http://www.resdc.cn | 1000m × 1000m | |
| **GDP** | Raster | Resource and Environment Science and Data Center http://www.resdc.cn | 1000m × 1000m | |
| **Digital Elevation Model (DEM)** | Raster | Geospatial Data Cloud, http://www.gscloud.cn | 30m × 30m | |
| **Population Density** | Raster | Resource and Environment Science and Data Center http://www.resdc.cn | 1000m × 1000m | |

**Table 2. Land use intensity classes standard.**

| Name | Unused land classes | Forest, grassland and waters classes | Agriculture classes | Building,village classes |
|---|---|---|---|---|
| **Land use type** | Unused land | Forest, grassland and waters | Cropland,garden | Towns, settlements, industrial and mining land, transportation land |
| **Classes index** | 1 | 2 | 3 | 4 |

where, $L$ was the land use intensity index for the study area. $J_i$ was the land use intensity classification index for category $i$. $A_i$ was the percentage of land use intensity area for category $i$. $N$ is the standard classification index for land use intensity.

**2.4.4. Habitat quality assessment.** The InVEST model presupposes land use types and habitat quality, which directly influence the suitability of biodiversity richness for habitats [33]. The size of the Habitat Quality Index (HQI) first required a calculation of the degree of habitat degradation. The formula is as follows:

$$D_{xj} = \sum_{i=1}^{R} \sum_{y=1}^{Y_r} \left( \frac{W_r}{\sum_{r=1}^{R} W_r} r_y i_{rxy} \beta_x S_{jr} \right)$$ (3)

$$i_{rxy} = 1 - \left( \frac{d_{xy}}{d_{rmax}} \right) \text{ (Linear recession)}$$ (4)

$$i_{rxy} = exp\left[ -\left( \frac{2.99}{d_{rmax}} \right) d_{xy} \right] \text{ (Index recession)}$$ (5)

where, $D_{xj}$ was the degree of habitat degradation. $Y_r$ was the total number of raster associated with threat factor $r$. $W_r$ was the weight of threat factor r, $W_r \in [0,1]$. $r_y$ was the coercion value of threat factor $r$ on raster $y$. R was the total number of threat factors. $i_{rxy}$ was the degree of threat to habitat raster $x$. $S_{jr}$ was the level of sensitivity to $r$ in type $j$ of land use. $i_{rxy}$ was the effect of $r$ on raster $x$ in raster $y$. $d_{xy}$ was the linear distance between raster $x$ and $y$. $d_{rmax}$ represented the maximum impact distance. On this basis, the final habitat quality was calculated by combining the indicators of landscape type sensitivity and external threat intensity [34]. The specific formula was as follows:

$$Q_{xj} = H_j \left( 1 - \frac{D_{xj}^z}{D_{xj}^z + K^z} \right)$$ (6)

where, $Q_{xi}$ is the habitat quality index for raster $x$ in land type $j$, $Q_{xi} \in [0,1]$. $H_j$ is the habitat suitability index of land type $j$. $D_{xj}^z$ is the disturbance level of raster $x$ in land type $j$. $k$ is the half-saturation constant, with a default value of 0.5. $z$ is the normalization constant, with a default value of 2.5.

In addition, paddy, dry land, buildings, and unused land were defined as threat factors. These threat factors were revised through the InVEST modeling handbook criteria and actual of the study area [35]. Combined with related studies in the Karst region, we determined the parameters of each landscape threat factor. Details were given in S2 and S3 of the supplementary material [36–39].

**2.4.5. Spatial autocorrelation analysis.** Moran's I index is a correlation that describes the correlation between neighboring units, i.e., the correlation between the spatial distribution characteristics of geographic phenomena and the degree of spatial agglomeration [40]. Based on the Arcgis10.2 and Geoda1.22 tools, spatial correlation analysis of habitat quality in karst mountainous areas was conducted. The spatial distribution of habitat quality was demonstrated by Lisa's cluster map. The spatial autocorrelation is calculated as follows [41]:

$$I = \frac{n \sum_{i=1}^{n} \sum_{j=1}^{n} w_{ij} \left( x_i - \bar{x} \right) \left( x_j - \bar{x} \right)}{\sum_{n=1}^{n} \left( x_i - \bar{x} \right) \left( \sum_{i=1}^{n} \sum_{j=1}^{n} w_{ij} \right)}$$ (7)

where, $I$ was Moran's index. $n$ was the number of cells in the study area. $x_i x_j$ were the values of cells $i$ and $j$, respectively. $\bar{x}$ was the mean value of spatial cells. $w_{ij}$ was the weight

matrix of spatial cells *i* and *j*. Moran's $I \in$ [-1,1]. when $I < 0$, there was a negative spatial correlation. when $I > 0$, there was a positive spatial correlation.

**2.4.6. Geographic detectors.** Geographic detectors were a statistical method for detecting the degree of spatio-temporal differentiation of variables and revealing the underlying drivers, which can be used to measure the relationships between habitat quality and various factors [42,43]. Q characterized the magnitude of influence of the detection factor. The formula was as follows:

$$Q = 1 - \frac{1}{N_\sigma^2} \sum_{h=1}^{L} N_h \sigma_h^2 \tag{8}$$

where, the value of *Q* was the explanatory power of the independent variable *X* on the dependent variable *Y*, $Q \in$ [0, 1]. The explanatory power increased with larger values. $h = (1,...,L)$ was the stratification of the independent variable *X* or the dependent variable *Y*. and N were the number of cells in stratum h and the entire region, respectively. and were the variance of the entire region in stratum h. The nature of the two-way interaction results can be found in the supplementary material S4.

This paper included eight impact factors to detect the influence of habitat quality. The factors considered include elevation, slope, land use intensity index, NDVI, annual precipitation, annual mean temperature, population density, and GDP as independent variables. We used the natural breakpoint method of the ArcGIS tool for the reclassification process. Geographic detectors were used to analyze the response relationship of different variables.

**2.4.7. Data processing and analysis.** Firstly, for analytical convenience, all data were resampled at a spatial resolution of 1000 × 1000 m. Spatio-temporal evolution of land use and landscape heterogeneity were analyzed using ArcGIS 10.2 and Fragstats 4.2.1 software. Secondly, the habitat quality index of the study area in different time periods was calculated using the InVEST model. Based on the county units, the spatial heterogeneity of regional habitat quality was analyzed using GeoDa software to elucidate the spatio-temporal evolution of habitat quality in different units. Finally, data on mean annual temperature, mean annual precipitation, slope, NDVI, GDP, and population density of different counties were extracted using ArcGIS to analyze the drivers of changes in habitat quality using the Geodetector tool.

## 3. Results

### 3.1. Land-use transfers and landscape pattern changes

**3.1.1. Land-use transfers.** The landscape types in the karst region changed dramatically between 1990 and 2020. Cropland, forest, and grassland are the main landscape types in the karst region (Fig. 3a). Among them, the building land area increased radially from the center to the periphery. The building land and waters showed a steady upward trend, increasing by 1869.32 km² and 786.39 km², respectively, compared with the early stage of the study (Fig. 3b). This is a result of the urbanization process, which has increased the level of migration. As of 2000, there was a significant downward trend in forest area, with a shift towards arable land and grassland. This period was negatively influenced by the region's limited arable land resources, increasing population density, and traditional agricultural culture. Between 2000 and 2010, forest cover showed a significant upward trend, with large areas of cropland and forest being converted to forest. During this period, China began large-scale projects to combat rocky desertification and return farmland to forest, in order to achieve the goal of ecological environmental restoration. After 2010, forest land and cropland showed a downward trend, with a shift to construction land and waters. During the study period, the

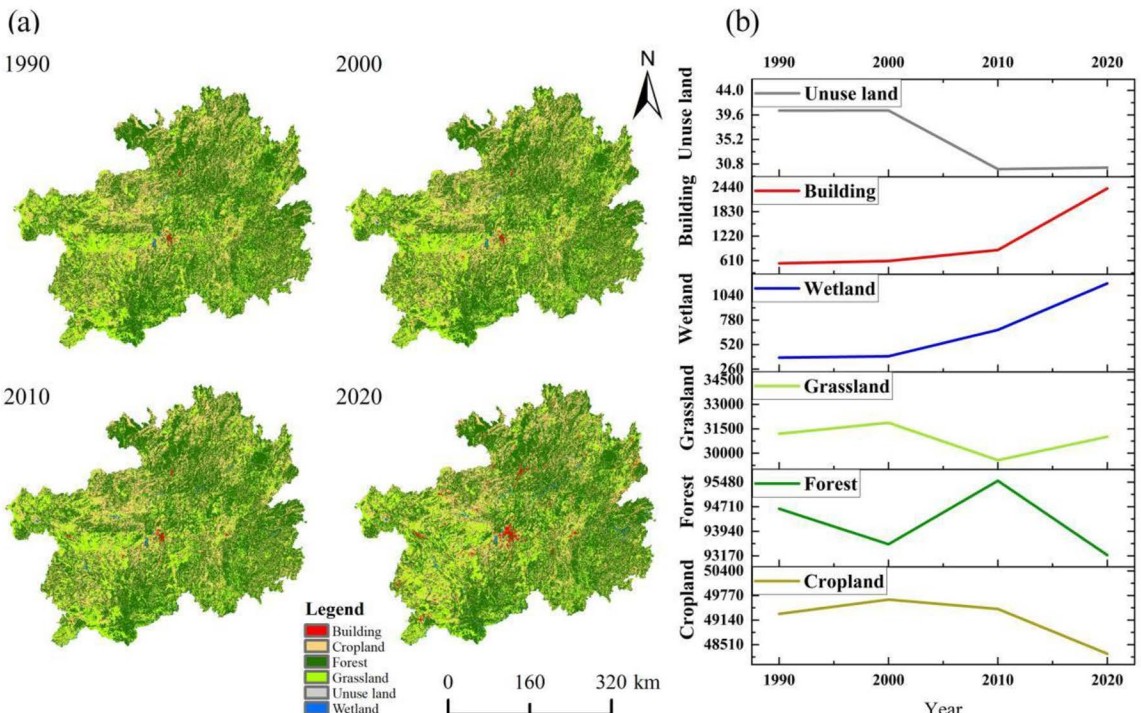

**Fig 3. Spato-temporal changes in land use types.**

main land categories decreased at different rates, with arable land decreasing by 1019.79 km², forest land by 1454.08 km², and grassland by 182.78 km². The results show that the arable land, forest land, and grassland have changed drastically, and the urbanization process has pushed the construction land and wetland area to expand rapidly, and the land use shift has been greatly influenced by human activities. This is an important feature and driving force of landscape pattern change in karst areas during this period.

**3.1.2. Landscape pattern changes.** The land use transfer matrix clearly shows the transfer of different landscapes within the karst region (Table 3). During the study period, cropland is mainly transformed into forest, grassland, and building land, with transformed areas of 3133.39 km², 1193.72 km², and 1121.03 km², respectively. Forests are mainly transformed into grassland, cropland, and building land, with transformation areas of 4599.17 km², 3137.96 km² and 409.24 km², respectively. Grassland was mainly transformed into forest and additional grassland, with a transformation areas of 3945.24 km² and 1515.03 km², respectively. Overall, cropland, forest, and grassland in the karst region have undergone drastic land type transformation, with the remarkable characteristics of three land types transformed into construction land.

**3.1.3. Analyzing change in landscape pattern indices.** There is significant variability in the pattern indices of different landscape types in karst. The type-level analysis showed that the number of patches (NP), landscape shape index (LSI), mean patch area (AREA_MN), and cohesion index (COHESION) for cropland and grassland are much higher than those of the other four land types (Fig. 4). Cropland and grassland are the dominant landscapes concentrated in the karst region during the study period, and the landscape indices for the different types are in a relatively flat trend. The AREA_MN for forest shows a fluctuating downward trend, indicating an increase in landscape fragmentation. During 1990-2010, the

**Table 3. Land use transfer matrix 1990-2020 (km²).**

| 1990 | 2020 | | | | | | |
|---|---|---|---|---|---|---|---|
| | Cropland | Forest | Grassland | Waters | Building | Unused land | Amount of change |
| Cropland | 43591.08 | 3133.39 | 1193.72 | 254.81 | 1121.03 | 0.76 | 5703.71 |
| Forest | 3137.96 | 86077.76 | 4599.17 | 409.24 | 412.35 | 2.51 | 8561.23 |
| Grassland | 1515.03 | 3945.24 | 25200.50 | 157.01 | 372.51 | 1.24 | 5991.03 |
| Waters | 13.35 | 16.08 | 7.20 | 343.80 | 1.62 | 0.03 | 38.28 |
| Building | 17.66 | 11.12 | 7.30 | 3.72 | 501.80 | 0.01 | 39.81 |
| Unused land | 1.64 | 8.82 | 2.60 | 0.10 | 1.61 | 25.60 | 14.77 |
| Amount of change | 4685.64 | 7114.65 | 5809.99 | 824.88 | 1909.12 | 4.55 | 20348.83 |

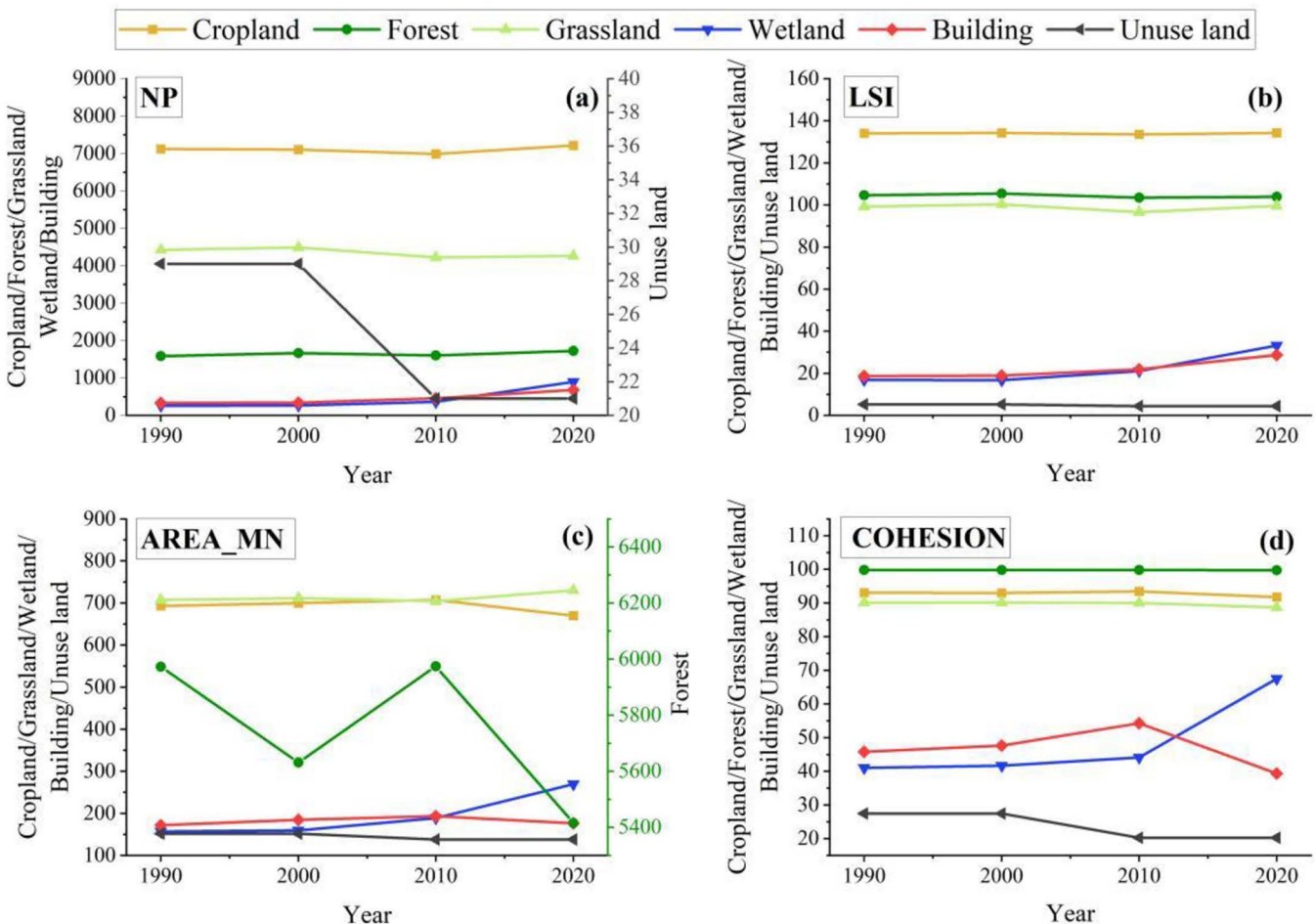

**Fig 4. Type level landscape index changes in 1990-2020.**

NP, LSI, AREA_MN, and COHESION of building land and unused land show an increasing trend, indicating that while the area increased the shape tended to be complicated, reflecting the agglomeration and expansion of built-up land. During 2010-2020, the AREA _MN and COHESION showed an abnormal decline, which is the opposite trend to that of waters, indicating a gradual fragmentation of building land.

In addition, the results of the analysis at the landscape level show that NP and the Fragmentation Index (FN) in the karst region showed a fluctuating upward trend, indicating an increase in landscape fragmentation and heterogeneity across the region as a whole (Table 4). On the contrary, AREA_MN and the Contagion Index (CONTAG) show a fluctuating downward trend, indicating a decrease in the connectivity of the main landscape types within the karst region. Meanwhile, the decrease of AREA_MN further proves the authenticity of the increase in NP. Shannon's Diversity Index (SHDI) shows a continuous upward trend, indicating that the area difference of different land types decreases the distribution of regional components is balanced, and the landscape heterogeneity gradually increases.

## 3.2. Analysis of spatial and temporal patterns of habitat quality

**3.2.1. Temporal trends and spatial patterns.** The InVEST model analysis shows the spatial and temporal evolution differences of the habitat quality indices within the karst region. The results of the temporal trend analysis show that the mean values of habitat quality in different years from 1900 to 2020 were 0.7751, 0.7719, 0.7626, and 0.7409, respectively (Table 5). The overall habitat quality shows a decreasing trend. Using the equidistant breakpoint method of ArcGIS software, we categorize the habitat quality of the area into five classes containing low (I), lower (II), moderate (III), higher (IV), and high (V). Of these, II and V are the dominant types of habitat quality indices, together accounting for more than 90% of the total. The current decline in class V is 5.4 times that of class II. This indicates the emergence of two more extreme ecological environments in the karst region. One is Class II, which is threatened by the spread of rocky desertification, while the other, with its unique geomorphological conditions, has a high-quality ecological environment similar to a primary forest. During the study period, II and V are transformed into classes I, III, and VI, and the three classes increase by 1.26%, 0.53%, and 3.21%, respectively. The overall habitat quality of the karst region has shown a downward trend over the last 30 years.

The results of the spatial distribution pattern show that the habitat quality index in the study area has an overall spatial distribution pattern of high in the periphery and low in the center (Fig 5). Grade I is more concentrated in the center of the area, and their proportion of area is increasing. The other classes were evenly distributed throughout the study area. The spatial distribution of the habitat quality index and land use change is very similar. It shows that the urbanization process is destroying habitat integrity and increasing regional landscape fragmentation and heterogeneity. Although the series of treatment projects have made a major contribution to ecological restoration, the effectiveness of treatment is difficult to consolidate.

In summary, we used the zoning statistics tool of the ArcGIS software to obtain a distribution map of changes in habitat quality from 1990 to 2020 (Fig 6). The results show that habitat quality within the county unit shows significant variability over time. From 1990 to 2000, there was a slight increase in habitat quality in the western region, while a slight decrease was observed in the east. In particular, the habitat quality index decreased in the northeastern

**Table 4. Pattern indices of landscape levels.**

| Year | NP | AREA_MN (km²) | CONTAG (%) | SHDI | FN |
|------|------|------|------|------|------|
| 1990 | 13763.00 | 1283.20 | 44.36 | 1.03 | 0.00078 |
| 2000 | 13905.00 | 1270.10 | 43.95 | 1.04 | 0.00079 |
| 2010 | 13665.00 | 1292.41 | 44.15 | 1.04 | 0.00077 |
| 2020 | 14810.00 | 1192.41 | 41.32 | 1.09 | 0.00084 |

**Table 5. Proportion and mean of each class of habitat quality.**

| Class | Classification range | Area percentage (%) | | | |
|---|---|---|---|---|---|
| | | 1990 | 2000 | 2010 | 2020 |
| I | 0-0.2 | 0.34 | 0.38 | 0.62 | 1.60 |
| II | 0.2-0.4 | 27.97 | 28.10 | 27.95 | 27.19 |
| III | 0.4-0.6 | 0.12 | 0.13 | 0.28 | 0.65 |
| IV | 0.6-0.8 | 0.83 | 1.02 | 1.87 | 4.04 |
| V | 0.8-1 | 70.74 | 70.19 | 69.29 | 66.35 |

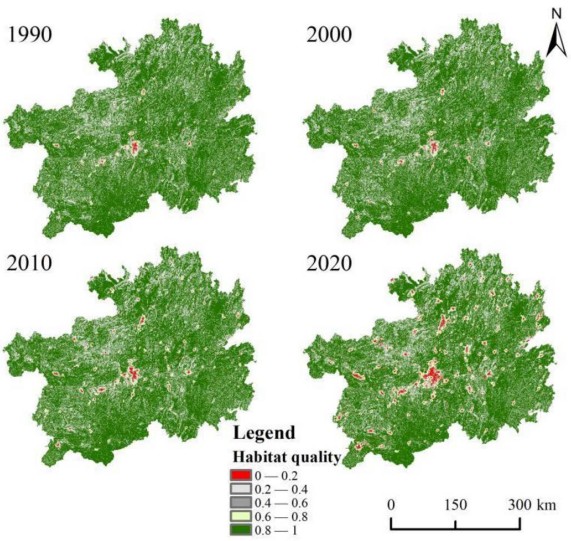

**Fig 5. Habitat quality in the study area from 1990 to 2020.**

counties of Yinjiang, Dejiang, and Yanhe. From 2000 to 2010, the Habitat Quality Index (HQI) in the study area showed declining in the center and a sporadic increase in the periphery. From 2010 to 2020, the radial decline in habitat quality from the center to the periphery slowly decreases. The overall habitat quality Index has shown a slight downward trend over the past 30 years, similar to the third phase. Regional natural habitats are increasingly threatened by loss, which inevitably leads to increased habitat fragmentation and heterogeneity.

**3.2.2. Habitat quality autocorrelation analysis.** The results of the global spatial autocorrelation analysis show that the global Moran'I values for habitat quality in the karst region from 1990 to 2020 are 0.452, 0.453, 0.474, and 0.504, respectively (P < 0.05). The spatial distribution of overall regional habitat quality has a more significant spatial positive correlation and spatial clustering. The global Moran'I value shows an increasing trend, indicating that the overall spatial clustering is increasing.

Localized spatial autocorrelation analysis based on county units shows that habitat quality correlations are not significant (NS) in most parts of the karst (Fig. 7). The "H-H" clustered areas are mainly distributed in the east and south of the study area and have showed an increasing trend over time. Compared to the previous two points, the "H-H" agglomeration in 2010 added Tianzhu County, Jinping County, Liping County, and Dushan County. In addition, Libo County was added in 2020. The overall habitat quality in the eastern and southern

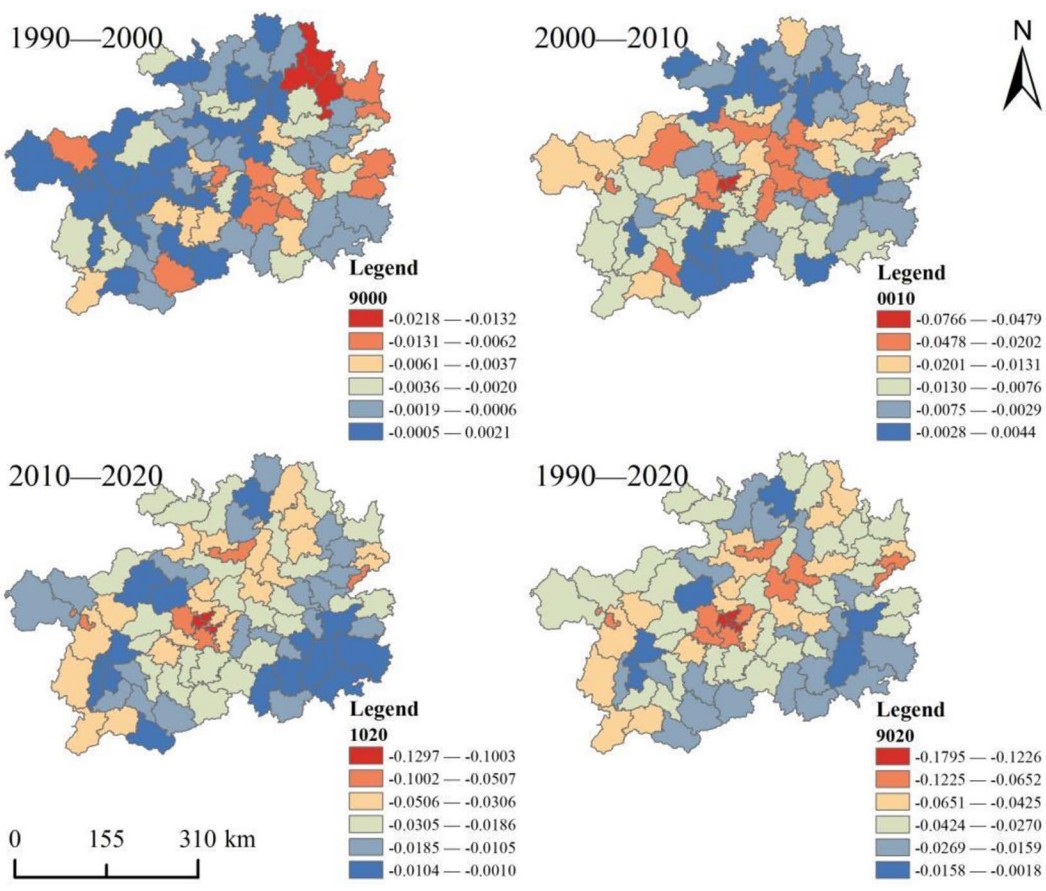

**Fig 6. Spatial distribution of changes in habitat quality in the study area.**

regions shows a favorable development. The "H-L" and "L-H" agglomerations do not show a significant distribution, except for Wudang District, which is located in the central urban agglomeration. The "L-L" agglomeration area is mainly located in the seven county-level regions of Yunyan, Huaxi, and Nanming districts in the central part of the study area, and there is no significant trend of change.

## 3.3. Drivers of change in habitat quality

**3.3.1. Factor detection analysis.** There is significant variability in the driving of habitat quality by factors of natural (temperature, precipitation, elevation, and NDVI) and anthropogenic (land use intensity, population density, and GDP) in the karst region (Fig 8). The results of factor detection show that the influence of slope and NDVI among natural factors is much higher than other factors. Slope and NDVI show a continuous upward trend, implying that reduced anthropogenic accessibility and increased vegetation cover became the characteristic features of high habitat quality during the period, with the largest increase in NDVI being 0.13. The influence of population density on habitat quality among the economic factors shows a fluctuating upward trend and had the most explanatory power in this result with an increase of 0.204. Land use intensity shows a fluctuating upward trend, which is consistent with the spatial and temporal evolution of habitat quality. But, neither natural nor economic factors can influence habitat quality independently, and there is a close relationship

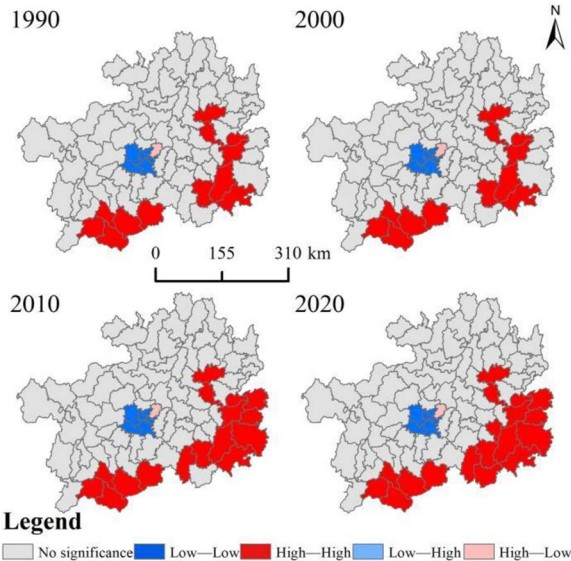

**Fig 7. Study area habitat quality LISA clustering map.**

between them. Therefore, there is an urgent need to investigate the effects of interactions between factors on habitat quality.

**3.3.2. Interaction detection.** There are significant differences in the effects of interactions between factors on habitat quality in karst areas (Fig 9). The total types all showed either two-factor or non-linear improvement. This reflects the fact that the interaction of different factors has the property of increasing explanatory power and that any two factors interact more than the effect of a single factor. The natural factor slope interacted more with the other factors, and it had an effect of more than 48% on the spatial differentiation of habitat quality. The strongest effects of slope and land use intensity interactions on the spatial and temporal evolution of habitat quality were observed from 2000 onwards. Thereafter, the strongest effects shifted to slope and population density. The influence factors were greater than 0.81 for all four time periods analyzed above. The natural factor interactions show a fluctuating upward trend over time, with the climate factor (temperature, precipitation) at a low level of interaction with other factors. The anthropogenic factors of land use intensity and population density have strong interactions with other factors, and their explanatory power for the spatial and temporal evolution of habitat quality exceeded 49%. Overall, slope, land use intensity, and population density are the dominant factors influencing the spatial and temporal evolution of habitat quality in karst areas.

## 4. Discussion

### 4.1. Landscape patterns and changes in their indices

Global landscape patterns have been altered to varying degrees by the interaction of multiple factors, with climate, land use, and population density being the dominant drivers of change [44]. However, the special binary three-dimensional geomorphic structure of the karst region area causes many problems, such as population expansion, rocky desertification expansion, and biodiversity decline [45]. At a certain time scale, there may be differences in the global landscape pattern change trend. The significant fluctuating trend of "decreasing-increasing-decreasing" was observed in the forest. This trend is partially different from the results of Wei

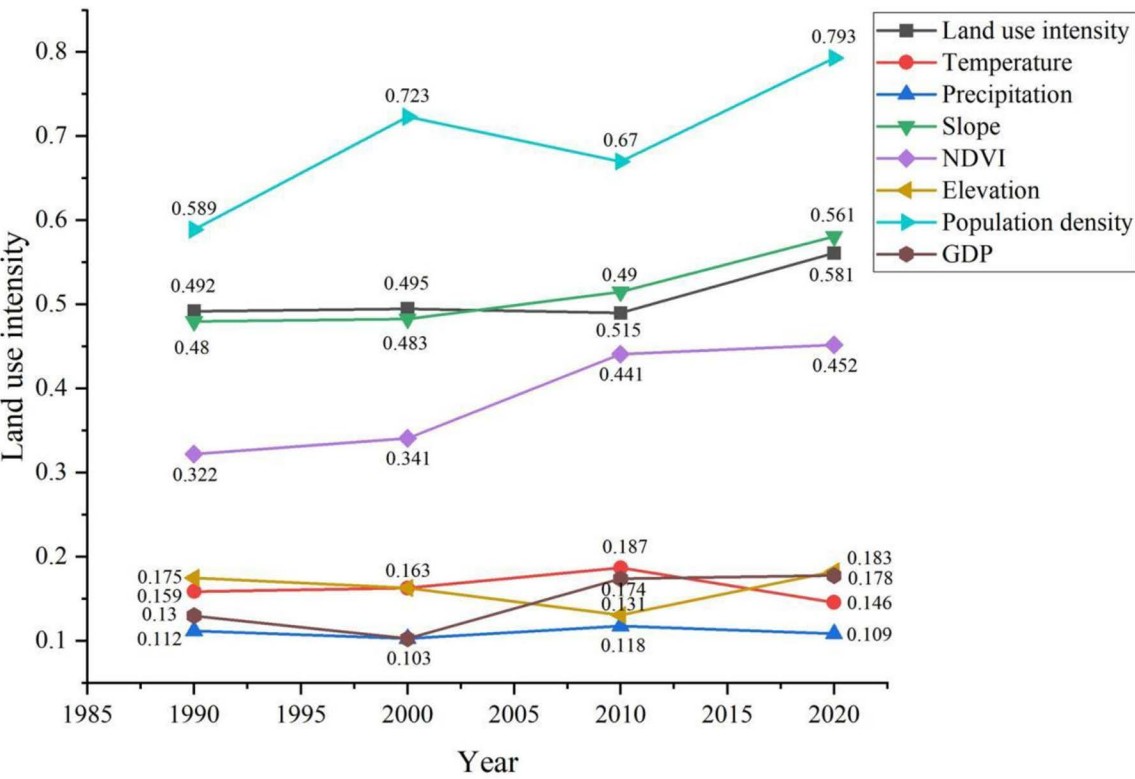

**Fig 8. Detection results of driving factors in karst areas.**

et al. [46]. From 2000 to 2010, the large-scale implementation of the Grain for Green Project (GGP) not only turned the South China Karst into a global greening hotspot, but also effectively halted land degradation, slowed biodiversity decline, and curbed the expansion of rocky desertification [47]. However, since 2010, the forest landscape has been gradually degraded to varying degrees. After the team's field research and observation, it was found that the growth and development of large areas of forest had stagnated, and ecosystem functions were seriously lagging. In this regard, we tried to find a solution to consolidate the effectiveness of rocky desertification management from the perspective of the landscape pattern index. The results of the study showed that the urbanization process changes the patch size of the forest landscape, leading to the fragmentation and heterogeneity of the overall landscape pattern in the region. This is in contrast to the findings of Wu et al. [37], which essentially stated that a series of ecological management projects did not significantly improve the status quo of fragmented landscapes, shallow soils, and monolithic ecosystem structures. In summary, changes in landscape patterns and indices directly altered the direction of spatial and temporal evolution of habitat quality.

## 4.2. Spato-temporal variation in habitat quality

Understanding the interaction between habitat quality with landscape patterns, and index changes is essential for effective ecological management. Habitat quality indices in karst areas have shown a decreasing trend of change over the last 30 decades and are dominated by habitat quality levels II and V. This trend of change is similar to the findings of Chen et al. [38] and Xie et al. [48]. The main reasons were soil and water leakage, single secondary vegetation

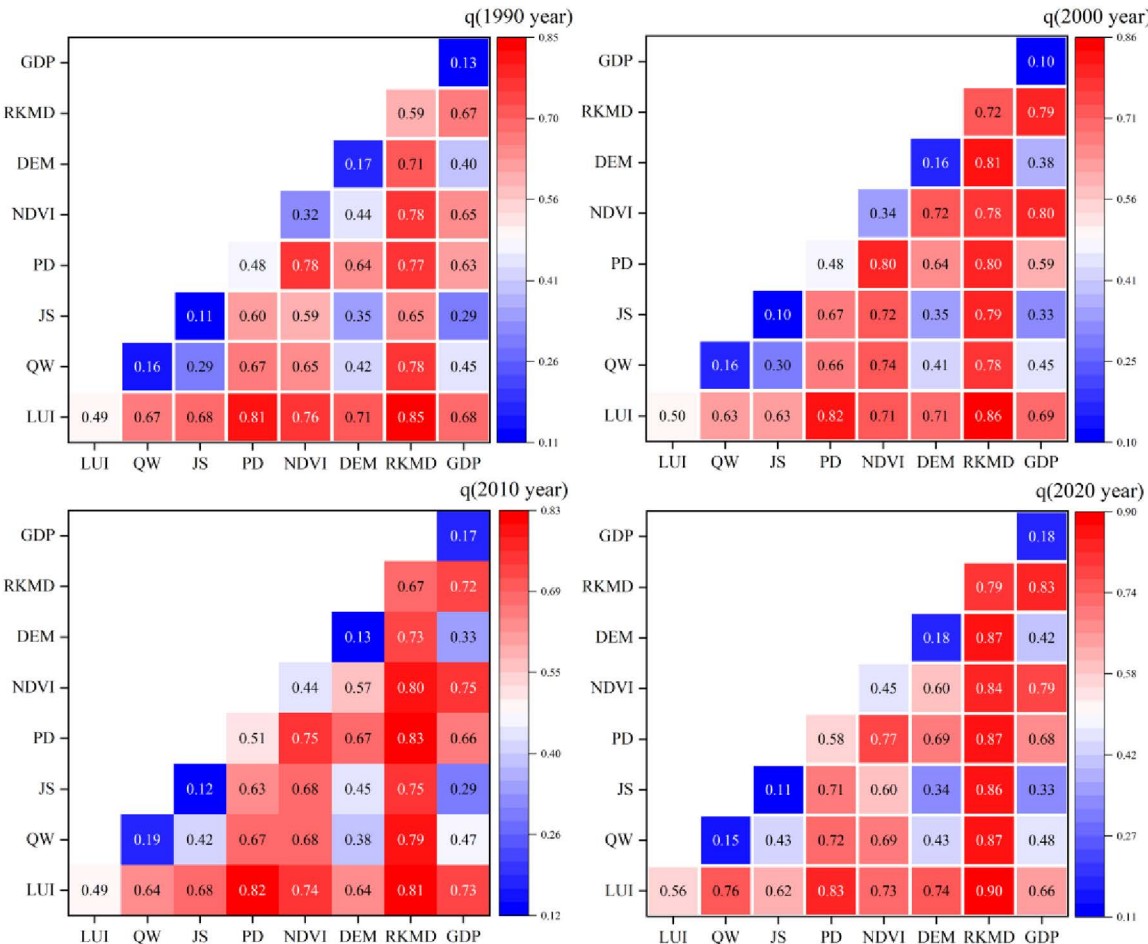

**Fig 9.  Interactive detection of habitat 1uality in the karst region 1990-2020.**

structure and biodiversity decline, and landscape fragmentation, which reduced the positive role of ecological corridors. In addition, the high spatial heterogeneity of the karst region caused extreme water and heat conditions in many microhabitats, ultimately resulting in a decline in ecosystem service capacity.

Meanwhile, using the ArcGIS spatial analysis tool, it was found that the habitat quality index showed a spatial distribution pattern of high periphery and low center, especially the class I was significantly concentrated in the center of the study area. The distribution pattern is similar to the results of Dong et al. [10] for the Yellow River Basin and Mengist et al. [49] for the Ethiopian Forest Reserve. The main reasons are that the large-scale urbanization process has seriously disturbed the primary succession of vegetation communities, and the area is characterized by the nature of high ecological vulnerability and low ecological carrying capacity. This leads to fragmentation, reduction, or loss of native plant and animal habitats, and the central area faces severe ecological degradation [50]. However, with the implementation of China's Rocky Desertification Control Project and the Targeted Poverty Alleviation (TPA) policy, poor people in remote mountainous areas have migrated to the periphery of cities [51]. On the contrary, the reduction of inappropriate anthropogenic economic activities in the surrounding area has improved habitat quality. The result of the spatial autocorrelation analysis is that the global Moran'I value shows an increasing trend. It indicates that the spatial

distribution of regional habitat quality is characterized by a significant increase in clustering. Among them, the area of built-up land in the central region increased and habitat connectivity weakened. In contrast, forests, grasslands, and water bodies, as well as good natural backgrounds, and the decrease in population density in the southeast, effectively improved the level of habitat quality [52]. However, the rate of recovery of habitat quality in neighboring areas remained at a low level (Fig 6). There is a lag between regional ecological processes and landscape pattern changes, which directly leads to the asynchrony between ecosystem functions and landscape pattern changes [53]. In the future, it is necessary to continue to strengthen the scientific promotion of ecological restoration projects and multi-scale vegetation ecological restoration initiatives to promote the overall habitat quality of karst regions.

## 4.3. Differences in drivers of habitat quality

Geographic detector methods have been widely used in studies investigating the drivers of habitat quality. Factor probe analysis showed that land use intensity, slope, and NDVI had a much higher influence than other factors. Both land use intensity and population density had a higher than 0.49 on habitat quality. This was similar to the findings of Chen et al.[54] in the western Tien Shan fruit forest area and Fan et al.[55] in the Hong River basin. Human activities are the main factors that change the quality of habitats. In addition, land use and population density are also triggering factors for the expansion of rocky desertification in karst areas in southern China, and the trend of changes in desertification rank over the past 35 years has been weakening-exacerbating-weakening [56]. Therefore, more attention needs to be paid to the interactive relationship between rock desertification and human activities in karst areas. However, the effects of each factor on habitat quality were not independent. Through interaction detection analysis, we found that all types of factors showed two-way and non-linear improvement. The interaction of land use intensity and population density with other factors had the greatest explanatory power for habitat quality in karst areas. This was at variance with the findings of Wu et al. [57]. The main reasons are the limited arable land resources in karst areas, the concentration of poor people, and the expansion of rocky desertification. It has changed the landscape pattern of land use and the spatial and temporal evolution trend of habitat quality. Although the potential risk of habitat degradation still exists, the shrinking population continues to reduce land pressure and disturbance intensity [58]. The future prospects for habitat quality and biodiversity restoration remain promising.

## 4.4. Management suggestions and limitations

**4.4.1. Management suggestions.** We analyze key drivers based on trends in karst landscape pattern indices and the spatial and temporal heterogeneity of habitat quality. Optimized landscape configuration is an important tool for the provision of multiple services and thus ecosystem management [59]. This contributes to the achievement of the 15th Sustainable Development Goal (SDG) in ecologically fragile karst areas. Land use and population density in fragile ecological zones are major factors exacerbating landscape fragmentation and undermining habitat integrity. With the accelerated processes of industrialization, urbanization, and agricultural modernization, the current land improvement program is more concerned with intensifying services to meet individual human needs, often neglecting the balanced development of the 'three life functions' of the rural landscape as a whole [60]. Therefore, we believe that different functional zones should be delineated according to the actual situation of regional vegetation, water and heat conditions, soil fertility, etc [61,62]. For example, the ecological protection zone for natural growth after afforestation in barren mountains, the production activity zone for the management of

agroforestry complexes, and the ecological recreation zone for the creation of urban green parks. To restore biodiversity and habitat integrity, as well as the ecological environment of karst areas.

**4.4.2. Limitations.** Although the InVEST model provides a framework for rapid, large-scale theoretical assessments of habitat quality. However, its underlying assessment parameters are based on empirical values specifically for assessing habitat quality in karst areas [54]. The InVEST model has the potential for the combined effects of multiple threat factors can be greater than those of a single factor [63]. In addition, karst areas suffer from many problems such as rockfalls, landslides, and fires, which may lead to inaccurate assessment results of habitat quality. In the future, we will obtain accurate rocky desertification information through the center of gravity model and a basic error matrix to study the interaction relationship and driving force of habitat quality, landscape pattern, and rocky desertification class. In-depth, multi-parameter, multi-scale, multi-scenario, and cross-integrated habitat quality studies are needed.

## 5. Conclusions

As one of the three most ecologically fragile regions in the world, the study of habitat quality, and its drivers in karst regions has been neglected. We systematically analyzed the spatial and temporal evolution of landscape patterns, habitat quality and its drivers based on land use data using the habitat quality plate of the InVEST model and Geographic Detectors methods. The study shows that: (i) The main types of forest, arable land, and grassland in the region have undergone drastic changes, with conversion to built-up land and water areas dominating. (ii) Regional landscape patches tend to be fragmented, with high complexity, diversity, and spatial heterogeneity. (iii) Regional habitat quality is dominated by classes II and V and shows a decreasing trend year by year, and its spatial distribution pattern is "high in the periphery and low in the center". (iv) Land use intensity and population density are the main drivers of the spatial and temporal evolution of habitat quality. The key finding is that habitat integrity and biodiversity are seriously threatened and that traditional large-scale afforestation projects have not been able to stem the loss of ecosystem multifunctionality and landscape fragmentation. In this context, based on the balanced development of ecological and economic benefits, we propose three spatial planning recommendations: restoration of natural afforestation, mixed agriculture, and forestry in line with management, and green parks. Future research should focus on the relationship between the environmental evolution patterns of rocky desertification and the response of habitat quality in different landscape types, and the construction of ecological corridors to restore landscape integrity. The results of this research will help to provide a scientific basis for decision-making on key functional areas and spatial planning.

## Supporting information

**S1 Table. Landscape index and its ecological significance.**
(DOCX)

**S2 Table. Habitat threat factors.**
(DOCX)

**S3 Table. Habitat suitability and threat factors sensitivity of the landscape types in the Guizhou province.**
(DOCX)

**S4 Table. Result types of two-factor interaction.**
(DOCX)

## Acknowledgments

We appreciate the anonymous reviewers for their invaluable comments and suggestions on this paper.

## Author contributions

**Conceptualization:** Chao Ma.

**Data curation:** Huituo Yang, Zhi Yan.

**Funding acquisition:** Chao Ma.

**Methodology:** Chao Ma.

**Software:** Huituo Yang, Zhi Yan.

**Supervision:** Huituo Yang, Zhi Yan.

**Writing – original draft:** Chao Ma.

**Writing – review & editing:** Huituo Yang, Zhi Yan.

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
