## [Decision Letter · Decision Letter 0]

15 Jul 2024

PONE-D-24-22242Spatio-temporal evolution of habitat quality and its influencing factors in karst areas based on the InVEST modelPLOS ONE

Dear Dr. Ma,

Thank you for submitting your manuscript to PLOS ONE. After careful consideration, we feel that it has merit but does not fully meet PLOS ONE’s publication criteria as it currently stands. Therefore, we invite you to submit a revised version of the manuscript that addresses the points raised during the review process.

We look forward to receiving your revised manuscript.

Kind regards,

Chun Liu

Academic Editor

PLOS ONE

 [This research was funded by Guiyang science and technology plan project (Zhuke contract [2021] No.43-17.

Guiyang University introduced talents to start the funds for scientific research project GYU-KY- [2023].].  

3. PLOS requires an ORCID iD for the corresponding author in Editorial Manager on papers submitted after December 6th, 2016. Please ensure that you have an ORCID iD and that it is validated in Editorial Manager. To do this, go to ‘Update my Information’ (in the upper left-hand corner of the main menu), and click on the Fetch/Validate link next to the ORCID field. This will take you to the ORCID site and allow you to create a new iD or authenticate a pre-existing iD in Editorial Manager. Please see the following video for instructions on linking an ORCID iD to your Editorial Manager account: https://www.youtube.com/watch?v=_xcclfuvtxQ".

4. Please include your tables as part of your main manuscript and remove the individual files. Please note that supplementary tables (should remain/ be uploaded) as separate ""supporting information"" files".

5. Please remove your figures from within your manuscript file, leaving only the individual TIFF/EPS image files, uploaded separately. These will be automatically included in the reviewers’ PDF.

Reviewers' comments:

Reviewer's Responses to Questions

**Comments to the Author**

1. Is the manuscript technically sound, and do the data support the conclusions?

Reviewer #1: Yes

Reviewer #2: Yes

Reviewer #3: Yes

2. Has the statistical analysis been performed appropriately and rigorously? 

Reviewer #1: Yes

Reviewer #2: Yes

Reviewer #3: Yes

3. Have the authors made all data underlying the findings in their manuscript fully available?

Reviewer #1: Yes

Reviewer #2: Yes

Reviewer #3: No

4. Is the manuscript presented in an intelligible fashion and written in standard English?

Reviewer #1: Yes

Reviewer #2: Yes

Reviewer #3: Yes

5. Review Comments to the Author

Reviewer #1: Suggestions and comments:

This paper primarily investigates the spatio-temporal variations in habitat quality in the karst regions of China and the driving factors behind these changes. The authors employed the InVEST model and GeoDetector method to analyze land use changes, the evolution of habitat quality from 1990 to 2020, and the primary driving factors. The methods used are sound and scientifically robust. The conclusions also provide strong scientific support for environmental protection. However, there are several areas that could be improved. The specific suggestions are as follows:

1. The literature review and theoretical background are not comprehensive enough. The paper lacks a thorough review and theoretical summary of existing related studies when introducing habitat quality assessment models and methods. It is recommended to include comparisons and analyses of other models (such as the SoLVES model) or methods (such as biophysical indicator assessments) to highlight the uniqueness and advantages of the InVEST model in this study. Given that InVEST is the primary research tool used in this paper and is widely utilized, it is not inherently innovative. Therefore, it is necessary to further emphasize the appropriateness and advantages of using this method in the current study.

2. The description of data sources and processing methods is unclear. In section 2.3, the paper does not provide detailed descriptions of the specific sources of data and the processing procedures. It is recommended to elaborate on how each type of data was obtained, the preprocessing steps involved, and the software and tools used. Additionally, there are two subsections labeled 2.3; this should be corrected.

3. The explanations and interpretations of the figures and tables are insufficient. While the figures (such as Figure 3 and Figure 5) in the paper present the research results, they lack detailed explanations and analyses of their contents. It is suggested to include explanations of key data points and trends in the figure descriptions, as well as how these data support the research conclusions. For example: "Figure 3 shows the trend of... from which we can observe...".

4. The conclusion section needs to be more explicit. The conclusion section (Section 5) of the paper provides a broad summary of the research findings but lacks specific policy recommendations and discussions on future research directions. It is suggested to clearly summarize the main findings of the study and propose specific policy recommendations and future research directions.

Reviewer #2: The article is an ordinary scientific research training article without any innovation. There are some obvious problems that need to be changed:

(1) In the article: "the InVEST-HQ model and Geographic Detectors methods."

The author should not create a non-existent method or method. The correct expression should be the habitat quality module in the InVEST model

(2) In the article, "Karst is a global greening hotspot", I think the karst area is a hotspot with very serious desertification

(3) "Manuscript" and Chinese characters appear in the article

(4) The description of the data in 2.3. Data source is extremely unclear

(5) For the translation of land use in the article, please refer to authoritative journals

(6) The management strategy in 4.4.1 cannot be obtained from your research, just write it according to the facts.

Reviewer #3: This manuscript is not innovative in terms of scientific issues, but it is of practical significance, especially for the regional development and construction of Guizhou Province. The writing of the manuscript is relatively standardized, the structure is clear, and the conclusions are well-founded. Combined with the journal policy of Plos One, I suggest the authors to accept it after minor revision. The following are the specific comments:

It is recommended that the authors remove the administrative boundary zoning of China or use the standard Chinese map issued by the Ministry of Natural Resources of China to produce an overview map of the study area.

The description of formulas needs to be written topically and with lowercase initial letters.

In Figure 6, "1990-2020", please make sure that this symbol is used correctly. Hyphens, minus signs, and long horizontal lines are different.

In Table 4, I suggest that the authors use "()" instead of "/" to denote units, because the latter has the meaning of division.

In addition, I also have a little suggestion, if the authors continue in this direction, they can consider the two directions of reducing the scale or improving the model, specifically: the impact of policy dominance on land use change and habitat quality in different regions, the use of the model to quantify the various influences, and the combination of the policy to predict the development trend of the region and the problems that will arise and to make recommendations and so on, you can refer to some of the Literature (doi.org/10.3390/rs14030527; doi.org/10.3390/rs15051385).

6. PLOS authors have the option to publish the peer review history of their article (what does this mean? ). If published, this will include your full peer review and any attached files.

**Do you want your identity to be public for this peer review?** For information about this choice, including consent withdrawal, please see our Privacy Policy .

Reviewer #1: No

Reviewer #2: No

Reviewer #3: No

---

## [Author Response · Author response to Decision Letter 0]

21 Aug 2024

Dear Editor:

Hello! We have completed all the revisions of the article and have responded to all of them individually in the order of the Q&A with Reviewer 1, 2, 3 and Editor 4. If there are any further questions, please contact us first and we look forward to hearing from you. Role of Funder statement：The funder of this study, Chao Ma, was responsible for the concept, the design, the writing and the financial support.

Yours sincerely,

Chao Ma

Response to Reviewer 1 Comments

Dear Reviewers,

Thank you very much for your comprehensive and constructive comments on our article, which were very helpful for us to improve this manuscript. The details of the issues are as follows：

Point 1: The literature review and theoretical background are not comprehensive enough. The paper lacks a thorough review and theoretical summary of existing related studies when introducing habitat quality assessment models and methods. It is recommended to include comparisons and analyses of other models (such as the SoLVES model) or methods (such as biophysical indicator assessments) to highlight the uniqueness and advantages of the InVEST model in this study. Given that InVEST is the primary research tool used in this paper and is widely utilized, it is not inherently innovative. Therefore, it is necessary to further emphasize the appropriateness and advantages of using this method in the current study.

Response 1: Thank you for this comment. We have reorganised the literature studies to highlight the advantages of the InVEST model by comparing different research models. We have made changes based on the suggestions. Detailed modifications are as follows: (Lines 45-69; Page 2):

Habitat quality is an important basis for maintaining ecosystem functions and providing ecosystem services. Current research has classified HQ assessment methods into three categories, namely field surveys, biophysical indicators and ecological modelling, based on differences in subject, content and scale [10]. Field survey assessment focuses on the biodiversity of regional populations and communities, as well as niche surveys. The method is only suitable for static assessments of small areas and short time series [11]. Biophysical indicators are based on remote sensing to provide an integrated assessment of ecological indicators, such as vegetation cover and net primary productivity (NPP). Boundary identification, data normalisation and indicator weighting processes are prerequisites of the approach, which lacks uniformity and generalisability [12]. However, the development of “3S” technology offers the possibility of assessing the spatial and temporal dynamics of ecosystems at multiple scales and long time series [13]. Examples include the Integrated Valuation and Trade-off of Ecosystem Services (InVEST) model and the Social Value of Ecosystem Services (SoLVES) model. Among them, the SoLVES model will integrate environmental and social values, but it is difficult and contradictory to quantify the aspects of values, attitudes and preferences of different stakeholder subjects [14,15]. However, the InVEST model, on the other hand, integrates habitat adaptations and levels of human disturbance and has the advantages of simplicity of use, easy access to data and strong spatial visualisation [16]. As a result, the InVEST model has been widely used to study HQ restoration strategies at large scales such as watersheds, provinces and globally [17-19], especially in studies based on the effects of land use change on HQ and biodiversity [20]. However, a single land use does not capture the extent to which human activities interfere with complex niche ecosystem processes. This is particularly true in karst areas, where environmental fragility, vegetation degradation and soil erosion are common phenomena. The landscape pattern index has the advantage of quantitatively analysing the degree of fragmentation, separation, biodiversity and disturbance of the spatial structure [21]. The evaluation method combining the InVEST model and the landscape pattern index can provide new ideas for the study of ecological status, spatial heterogeneity and human activities in karst areas.

Point 2: The description of data sources and processing methods is unclear. In section 2.3, the paper does not provide detailed descriptions of the specific sources of data and the processing procedures. It is recommended to elaborate on how each type of data was obtained, the preprocessing steps involved, and the software and tools used. Additionally, there are two subsections labeled 2.3; this should be corrected.

Response 2: Thank you for this comment. We have made changes based on the suggestions. Detailed modifications are as follows: (Lines 123; Page 4):

First, we placed the data sources in the body of the manuscript to facilitate a clear presentation of the data type, source, resolution, and time of acquisition.

Table 1. Data source

Name Type Data source Note Time

Land use/Land cover Raster Resource and Environment Science and Data Center

http://www.resdc.cn 30 m×30 m Accessed on 10 October 2023

Annual temperature Raster China Meteorological Data Center, http://data.cma.cn/ 1000 m×1000 m

Annual precipitation Raster China Meteorological Data Center, http://data.cma.cn/ 1000 m×1000 m

NDVI Raster Resource and Environment Science and Data Center

http://www.resdc.cn 1000 m×1000 m

GDP Raster Resource and Environment Science and Data Center

http://www.resdc.cn 1000 m×1000 m

Digital Elevation Model (DEM) Raster Geospatial Data Cloud,

http://www.gscloud.cn 30 m×30 m

Population Density Raster Resource and Environment Science and Data Center

http://www.resdc.cn 1000 m×1000 m

Secondly, we include relevant treatments and analysis processes in the main text. The specifics are as follows(Lines 193-201; Page 7):

2.4.7. Data processing and analysis

Firstly, for analytical convenience, all data were resampled at a spatial resolution of 1000 × 1000 m. Spatio-temporal evolution of land use and landscape heterogeneity were analysed using ArcGIS 10.2 and Fragstats 4.2.1 software. Secondly, the habitat quality index of the study area in different time periods was calculated using the InVEST model. Based on the county units, the spatial heterogeneity of regional habitat quality was analysed using Geoda software to elucidate the spatio-temporal evolution of habitat quality in different units. Finally, data on mean annual temperature, mean annual precipitation, slope, NDVI, GDP and population density of different counties were extracted using Arcgis to analyse the drivers of changes in habitat quality using the Geodetector tool.

Thirdly, we have amended the incorrect subsection headings and checked the full text. The specifics are as follows( Lines 125-192; Page 4-6):

2.4. Study methods

2.4.1. Land use transfer

2.4.2. Landscape pattern changes

2.4.3. Land use intensity

2.4.4. Habitat Quality Assessment

2.4.5. Spatial autocorrelation analysis

2.4.6. Geographic detectors

Point 3: The explanations and interpretations of the figures and tables are insufficient. While the figures (such as Figure 3 and Figure 5) in the paper present the research results, they lack detailed explanations and analyses of their contents. It is suggested to include explanations of key data points and trends in the figure descriptions, as well as how these data support the research conclusions. For example: "Figure 3 shows the trend of... from which we can observe...".

Response 3: Thank you for this comment. We have made changes based on the suggestions. Firstly, detailed modifications are as follows(Lines 209-222; Page 7):

This is a result of the urbanization process, which has increased the level of migration. As of 2000 , there was a significant downward trend in forest area, with a shift towards arable land and grassland. This period was negatively influenced by the region's limited arable land resources, increasing population density and traditional agricultural culture. Between 2000 and 2010, forest cover showed a significant upward trend, with large areas of cropland and forest being converted to forest. During this period, China began large-scale projects to combat rocky desertification and return farmland to forest, in order to achieve the goal of ecological environmental restoration. After 2010, forest land and cropland showed a downward trend, with a shift to construction land and waters. During the study period, the main land categories decreased at different rates, with arable land decreasing by 1019.79 km2, forest land by 1454.08 km2 and grassland by 182.78 km2. The results show that the arable land, forest land and grassland have changed drastically, and the urbanisation process has pushed the construction land and wetland area to expand rapidly, and the land use shift has been greatly influenced by human activities. This is an important feature and driving force of landscape pattern change in karst areas during this period.

Secondly, detailed modifications are as follows(Lines 260-281; Page 9-10):

The InVEST model analysis shows the spatial and temporal evolution differences of the habitat quality indices in the karst region. The results of temporal trend analysis show that the mean values of habitat quality in different years from 1900 to 2020 were 0.7751, 0.7719, 0.7626 and 0.74085, respectively (Table 5). The overall habitat quality shows a decreasing trend. Using the equidistant breakpoint method of ArcGIS software, we categorize the habitat quality of the area into five classes containing low (I), lower (II), moderate (III), higher (IV) and high (V). Of these, II and V are the dominant types of habitat quality indices, together accounting for more than 90% of the total. The current decline in class V is 5.4 times that of class II. This indicates the emergence of two more extreme ecological environments in the karst region. One is Class II, which is threatened by the spread of rocky desertification, while the other, with its unique geomorphological conditions, has a high-quality ecological environment similar to a primary forest. During the study period, II and V are transformed into classes I, III and VI, and the three classes increase by 1.26%, 0.53% and 3.21%, respectively. The overall habitat quality of the Karst region has shown a downward trend over the last 30 years.

The results of the spatial distribution pattern show that the habitat quality index in the study area has an overall spatial distribution pattern of high in the periphery and low in the center (Fig 5). Grade I is more concentrated in the center of the area, and the proportion of area is increasing. The other classes were evenly distributed throughout the study area. The spatial distribution of the habitat quality index and land use change is very similar. It shows that the urbanisation process is destroying habitat integrity and increasing regional landscape fragmentation and heterogeneity. Although the series of treatment projects have made a major contribution to ecological restoration, the effectiveness of treatment is difficult to consolidate.

Point 4: The conclusion section needs to be more explicit. The conclusion section (Section 5) of the paper provides a broad summary of the research findings but lacks specific policy recommendations and discussions on future research directions. It is suggested to clearly summarize the main findings of the study and propose specific policy recommendations and future research directions.

Response 4: Thank you for this comment. We summarise this section. Key points, management recommendations and future directions are articulated. Detailed modifications are as follows: (Lines 450-468; Page 14-15):

As one of the three most ecologically fragile regions in the world, the study of habitat quality and its drivers in karst regions has been neglected. We systematically analysed the spatial and temporal evolution of landscape patterns, habitat quality and its drivers based on land use data using the habitat quality plate and Geographic Detectors methods of the InVEST model. The study shows that: (i) The main types of forest, arable land and grassland in the region have undergone drastic changes, with conversion to built-up land and water areas dominating. (ii) Regional landscape patches tend to be fragmented, with high complexity, diversity and spatial heterogeneity. (iii) Regional habitat quality is dominated by classes II and V and shows a decreasing trend year by year, and its spatial distribution pattern is “high in the periphery and low in the centre”. (iv) Land use intensity and population density are the main drivers of the spatial and temporal evolution of habitat quality. The key finding is that habitat integrity and biodiversity are seriously threatened and that traditional large-scale afforestation projects have not been able to stem the loss of ecosystem multifunctionality and landscape fragmentation. In this context, based on the balanced development of ecological and economic benefits, we propose three spatial planning recommendations: restoration of natural afforestation, mixed agriculture and forestry in line with management, and green parks. Future research should focus on the relationship between the environmental evolution patterns of rocky desertification and the response of habitat quality in different landscape types, and the construction of ecological corridors to restore landscape integrity. The results of this research will help to provide a scientific basis for decision-making on key functional areas and spatial planning.

Response to Reviewer 2 Comments

Dear Reviewers,

Thank you very much for your comprehensive and constructive comments on our article, which were very helpful for us to improve this manuscript. The details of the issues are as follows：

Point 1: In the article: "the InVEST-HQ model and Geographic Detectors methods."The author should not create a non-existent method or method. The correct expression should be the habitat quality module in the InVEST model.

Response 1: Thank you for this comment. We have made changes based on the suggestions. Detailed modifications are as follows (Lines 450; Page 14):

“... using the habitat quality plate of the InVEST model ...”.

This is a misunderstanding caused by a misrepresentation on our part. We originally meant the habitat quality plate of the InVEST model.

Point 2: In the article, "Karst is a global greening hotspot", I think the karst area is a hotspot with very serious desertification

Response 2: Thank you for this comment. We have made changes based on the suggestions. Detailed modifications are as follows (Lines 447-448; Page 14):

“As one of the three most ecologically fragile regions in the world, the study of habitat quality and its drivers in karst regions has been neglected. We systematically analysed the spatial and temporal evolution of landscape patterns...”

A related commentary reports that the karst region of southern China is slowly turning green on satellite imagery[1]. However, in order to minimise the controversy over the presentation. We have deleted and amended this statement.

[1]Macias-Fauria, M. Satellite images show China going green[J]. Nature sustainability, 2018, 411-413.

Point 3: "Manuscript" and Chinese characters appear in the article

Response 3: Thank you for this comment. We have made changes based on the suggestions. Detailed modifications are as follows:

(1)We have removed the word “manuscript” from the text.

(2) For the emergent Chinese characters, we perform a spatial correlation analysis of county units in subsection 3.2.2 of this paper. For example, Tianzhu County, Jinping County, Liping County, Dushan County, Libo County,Yunyan, Huaxi and Nanming are not Chinese characters, they are names of county units. We apologize for this misunderstanding.

Point 4: The description of the data in 2.3. Data source is extremely unclear

Response 4: Thank you for this comment. We have made changes based on the suggestions. Detailed modifications are as follows (Lines 122; Page 4):

The source of the data, which we used to place in the Supplementary Materials, was a huge inconvenience for reviewers. We are now placing the data sources in the article to make it easier to read.

Table 1. Data

---

## [Decision Letter · Decision Letter 1]

18 Sep 2024

PONE-D-24-22242R1Spatio-temporal evolution of habitat quality and its influencing factors in karst areas based on the InVEST modelPLOS ONE

Dear Dr. Ma,

Thank you for submitting your manuscript to PLOS ONE. After careful consideration, we feel that it has merit but does not fully meet PLOS ONE’s publication criteria as it currently stands. Therefore, we invite you to submit a revised version of the manuscript that addresses the points raised during the review process.

We look forward to receiving your revised manuscript.

Kind regards,

Chun Liu

Academic Editor

PLOS ONE

Journal Requirements:

Reviewers' comments:

Reviewer's Responses to Questions

**Comments to the Author**

1. If the authors have adequately addressed your comments raised in a previous round of review and you feel that this manuscript is now acceptable for publication, you may indicate that here to bypass the “Comments to the Author” section, enter your conflict of interest statement in the “Confidential to Editor” section, and submit your "Accept" recommendation.

Reviewer #1: All comments have been addressed

Reviewer #2: All comments have been addressed

Reviewer #3: (No Response)

2. Is the manuscript technically sound, and do the data support the conclusions?

Reviewer #1: Yes

Reviewer #2: Yes

Reviewer #3: Yes

3. Has the statistical analysis been performed appropriately and rigorously? 

Reviewer #1: Yes

Reviewer #2: Yes

Reviewer #3: I Don't Know

4. Have the authors made all data underlying the findings in their manuscript fully available?

Reviewer #1: Yes

Reviewer #2: Yes

Reviewer #3: No

5. Is the manuscript presented in an intelligible fashion and written in standard English?

Reviewer #1: Yes

Reviewer #2: Yes

Reviewer #3: Yes

6. Review Comments to the Author

Reviewer #1: I have carefully reviewed the authors' responses and revisions based on the previous round of comments and found that the authors have made substantial improvements in several areas, including the supplementation of the literature review, refinement of the research methods, further explanation of the results, and deepening of policy strategies. These revisions have adequately addressed my previous concerns. I believe the paper is now suitable for acceptance.

Reviewer #2: Some of the language in the article needs further revision and polishing. I did not check all the references, and I noticed that some of them were not appropriate.

Reviewer #3: The main comments have been revised and formally the following details need to be improved to be publishable.

For Table 1, make sure the font is New Roman and not a mix.

Row 143, needs to be written top space (no indentation).

Table 3 needs to be revised and the land use transfer matrix can be referenced in the relevant literature.

Table 5, and also Table 6, the form of the table is a bit confusing.

7. PLOS authors have the option to publish the peer review history of their article (what does this mean? ). If published, this will include your full peer review and any attached files.

**Do you want your identity to be public for this peer review?** For information about this choice, including consent withdrawal, please see our Privacy Policy .

Reviewer #1: No

Reviewer #2: No

Reviewer #3: No

---

## [Author Response · Author response to Decision Letter 1]

14 Oct 2024

Dear Editors，

Greetings! We have made all the necessary changes to the article and have responded to each reviewer individually, addressing their questions in the order of questions 1, 2, and 3. If you have any further inquiries, please do not hesitate to contact us. We look forward to hearing from you.

Yours sincerely,

Chao Ma

Response to Reviewer 1 Comments

Dear Reviewers,

Thank you for recognizing our work. We will carefully review and correct the full text. Thank you for your hard work on this article.

Response to Reviewer 2 Comments

Dear Reviewers,

Thank you very much for your comprehensive and constructive comments on our article, which were very helpful for us to improve this manuscript. The details of the issues are as follows：

Point 1: Some of the language in the article needs further revision and polishing. I did not check all the references, and I noticed that some of them were not appropriate.

Response 1: Thank you for this comment. We attend English professionals to revise and embellish the language throughout the article. We checked references throughout and corrected inappropriate references. Detailed modifications are as follows: (Lines 495-661, Page 15-19):

2 Williams SE, Hobday AJ, Falconi L, Hero J, Holbrook NJ, Capon S, Bond NR, Ling SD, Hughes L. Research priorities for natural ecosystems in a changing global climate. Glob. Change. Bio. 2020; 26, 410-416.

7 Chen CZ, Ge XG, Sun L, Shao D, Ke WL. 9.Dramatic species loss · severe ecological overload · crossing the “earth boundary” · regional equity imbalances · “living on one planet” — interpreting The Earth Vitality Report 2014. Acta Ecologica Sinica, 2016; 36, 2779-2785.

8 Williams BA, Watson JEM, Butchart SHM, et al. A robust goal is needed for species in the Post‐2020 Global Biodiversity Framework. Conservation Letters, 2021; 14, e12778. https://doi.org/10.1111/conl.12778.

27 Han HQ, Liu Y, Gao HJ, Zhang YJ, Wang Z, Chen XQ, Tradeoffs and synergies between ecosystem services: A comparison of the karst and non-karst area. J Mt. Sci. 2020; 17, 1221–1234. https://doi.org/10.1007/s11629-019-5667-5.

59 IPBES W. Intergovernmental science-policy platform on biodiversity and ecosystem services. Summary for Policy Makers of the Global Assessment Report on Biodiversity and Ecosystem Services of the Intergovernmental Science-Policy Platform on Biodiversity and Ecosystem Services. IPBES Secretariat, Bonn, Germany, 2019.

Response to Reviewer 3 Comments

Dear Reviewers,

Thank you very much for your comprehensive and constructive comments on our article, which were very helpful for us to improve this manuscript. The details of the issues are as follows：

Point 1: For Table 1, make sure the font is New Roman and not a mix.

Response 1: Thank you for this comment. We have made changes based on the suggestions. We have changed the fonts in Table 1 uniformly to the Times New Roman font. Detailed modifications are as follows: (Lines 122, Page 4):

Table 1. Data source

Name Type Data source Note Time

Land use/Land cover Raster Resource and Environment Science and Data Center

http://www.resdc.cn 30m×30m Accessed on 10 October 2023

Annual temperature Raster China Meteorological Data Center, http://data.cma.cn/ 1000m×

1000m

Annual precipitation Raster China Meteorological Data Center, http://data.cma.cn/ 1000m×

1000m

NDVI Raster Resource and Environment Science and Data Center

http://www.resdc.cn 1000m×

1000m

GDP Raster Resource and Environment Science and Data Center

http://www.resdc.cn 1000m×

1000m

Digital Elevation Model (DEM) Raster Geospatial Data Cloud,

http://www.gscloud.cn 30m×30m

Population Density Raster Resource and Environment Science and Data Center

http://www.resdc.cn 1000m×

1000m

Point 2: Row 143, needs to be written top space (no indentation).

Response 2: We have changed line 143 to be written top space. We have checked the entire article and corrected every error. Detailed modifications are as follows:

(Lines 129, Page 5): “where, N is the unit of variation.”

(Lines 143, Page 5): “where, L was the land use intensity index for ...”.

(Lines 151, Page 6): “where, was the degree of habitat degradation.”

(Lines 158, Page 6): “where, is the habitat quality index..”.

(Lines 171, Page 6): “where, I was Moran's index.”

(Lines 180, Page 6): “where, the value of Q was the explanatory power...”.

Point 3: Table 3 needs to be revised and the land use transfer matrix can be referenced in the relevant literature.

Response 3: Thank you for this comment. We have made an addition to Table 3 which does contain. I apologize that this was an oversight in our work. Detailed modifications are as follows: (Lines 230, Page 8):

1990 2020

Cropland Forest Grassland Waters Building Unused

land Amount of change

Cropland 43591.08 3133.39 1193.72 254.81 1121.03 0.76 5703.71

Forest 3137.96 86077.76 4599.17 409.24 412.35 2.51 8561.23

Grassland 1515.03 3945.24 25200.50 157.01 372.51 1.24 5991.03

Waters 13.35 16.08 7.20 343.80 1.62 0.03 38.28

Building 17.66 11.12 7.30 3.72 501.80 0.01 39.81

Unused land 1.64 8.82 2.60 0.10 1.61 25.60 14.77

Amount of change 4685.64 7114.65 5809.99 824.88 1909.12 4.55 20348.83

We have modified Table 4 by referring to Liu et al [1].

[1] Liu JY, Ning J, Kuang WH, et al. Spatio-temporal patterns and characteristics of land-use change in China during 2010-2015. Acta Geographica Sinica, 2018, 73 (05): 789-802.

Point 4: Table 5, and also Table 6, the form of the table is a bit confusing.

Response 4: Thank you for this comment. We have modified Tables 5 and 6. Detailed modifications are as follows: (Lines 278, Page 10; Lines 324, Page 11):

Table 5. Proportion and mean of each class of habitat quality

Class Classification range Area percentage (%)

1990 2000 2010 2020

I 0-0.2 0.34 0.38 0.62 1.60

II 0.2-0.4 27.97 28.10 27.95 27.19

III 0.4-0.6 0.12 0.13 0.28 0.65

IV 0.6-0.8 0.83 1.02 1.87 4.04

V 0.8-1 70.74 70.19 69.29 66.35

We have changed one of the expressions in Table 6. We use a line graph to express it, which is more intuitive and clearer.

Fig 8. Detection results of driving factors in karst areas

---

## [Editor Report · Decision Letter 2]

6 Nov 2024

Spatio-temporal evolution of habitat quality and its influencing factors in karst areas based on the InVEST model

PONE-D-24-22242R2

Dear Dr. Ma,

We’re pleased to inform you that your manuscript has been judged scientifically suitable for publication and will be formally accepted for publication once it meets all outstanding technical requirements.

Kind regards,

Chun Liu

Academic Editor

PLOS ONE
---

## [Editor Report · Acceptance letter]

PONE-D-24-22242R2

PLOS ONE

Dear Dr. Ma,

I'm pleased to inform you that your manuscript has been deemed suitable for publication in PLOS ONE. Congratulations! Your manuscript is now being handed over to our production team.

Kind regards,

on behalf of

Dr. Chun Liu

Academic Editor

PLOS ONE